# Targeting Fatty Acid Amide Hydrolase Counteracts the Epithelial-to-Mesenchymal Transition in Keratinocyte-Derived Tumors

**DOI:** 10.3390/ijms242417379

**Published:** 2023-12-12

**Authors:** Daniela Kovacs, Enrica Flori, Emanuela Bastonini, Sarah Mosca, Emilia Migliano, Carlo Cota, Marco Zaccarini, Stefania Briganti, Giorgia Cardinali

**Affiliations:** 1Laboratory of Cutaneous Physiopathology and Integrated Center of Metabolomics Research, San Gallicano Dermatological Institute, IRCCS, 00144 Rome, Italy; daniela.kovacs@ifo.it (D.K.); enrica.flori@ifo.it (E.F.); emanuela.bastonini@ifo.it (E.B.); sarah.mosca@ifo.it (S.M.); stefania.briganti@ifo.it (S.B.); 2Department of Plastic and Reconstructive Surgery, San Gallicano Dermatological Institute, IRCCS, 00144 Rome, Italy; emilia.migliano@ifo.it; 3Genetic Research, Molecular Biology and Dermatopathology Unit, San Gallicano Dermatological Institute, IRCCS, 00144 Rome, Italy; carlo.cota@ifo.it (C.C.); marco.zaccarini@ifo.it (M.Z.)

**Keywords:** keratinocyte-derived skin cancers, epithelial-to-mesenchymal transition, endocannabinoids, FAAH inhibition, bioactive lipids

## Abstract

The endocannabinoid system regulates physiological processes, and the modulation of endogenous endocannabinoid (eCB) levels is an attractive tool to contrast the development of pathological skin conditions including cancers. Inhibiting FAAH (fatty acid amide hydrolase), the degradation enzyme of the endocannabinoid anandamide (AEA) leads to the increase in AEA levels, thus enhancing its biological effects. Here, we evaluated the anticancer property of the FAAH inhibitor URB597, investigating its potential to counteract epithelial-to-mesenchymal transition (EMT), a process crucially involved in tumor progression. The effects of the compound were determined in primary human keratinocytes, ex vivo skin explants, and the squamous carcinoma cell line A431. Our results demonstrate that URB597 is able to hinder the EMT process by downregulating mesenchymal markers and reducing migratory potential. These effects are associated with the dampening of the AKT/STAT3 signal pathways and reduced release of pro-inflammatory cytokines and tumorigenic lipid species. The ability of URB597 to contrast the EMT process provides insight into effective approaches that may also include the use of FAAH inhibitors for the treatment of skin cancers.

## 1. Introduction

The endocannabinoid system (ECS) is a complex signaling network including endogenous ligands known as endocannabinoids (eCBs), their receptors and transporters, and enzymes involved in eCB synthesis and degradation. The first eCBs identified in humans were anandamide (N-arachidonoylethanolamine, AEA) and 2-arachidonoylglicerol (2-AG), two bioactive lipids derived from arachidonic acid [1,2]. Later, other molecules were discovered and classified as endocannabinoid-like substances, such as palmitoylethanolamide (PEA) and oleoylethanolamide (OEA) [3]. eCBs are produced “on demand” as an adaptive response to cellular stress to re-establish cell homeostasis. They exert biological functions mainly through the activation of the cannabinoid receptors CB1 and CB2, which belong to the family of G-protein-coupled receptors (GPCRs) [4,5,6], and of peroxisome proliferator-activated nuclear receptors (PPARα and PPARγ). AEA and 2-AG are metabolized by fatty acid amide hydrolase (FAAH) and monoacylglycerol lipase (MAGL), releasing arachidonic acid (AA) and ethanolamine or glycerol, respectively. The endocannabinoid system is engaged in different physiological processes, thus representing an important homeostatic modulator. As a consequence, ECS deregulation is associated with several pathological conditions, including neurodegenerative disorders, cardiovascular disease, obesity, and cancer [7,8,9,10,11,12]. Targeting the endocannabinoid system as a potential anticancer strategy is an emerging interest. On the one hand, evidence from preclinical studies and clinical approaches using cannabinoid-based drugs demonstrated their efficacy in counteracting chemotherapy-induced nausea, vomiting, and pain [11,13,14,15]. On the other hand, in vitro and in vivo data revealed that eCBs exert anti-tumor effects, inhibiting cell proliferation, migration/invasiveness, angiogenesis, and epithelial-to-mesenchymal transition (EMT) while inducing apoptosis and autophagy in different cancer types [11,16,17,18].

The skin possesses a functional endocannabinoid system known to regulate physiological processes. Hence, the deregulation of ECS contributes to the development of a broad spectrum of cutaneous pathological conditions including pigmentary disorders, inflammatory diseases, and skin cancers [19]. eCBs exert anti-tumor effects in both melanoma and non-melanoma skin cancer cells [20,21,22,23,24,25,26]. The epidermal keratinocyte-derived cancer cutaneous squamous cell carcinoma (CSCC) is the second most common malignant disease in the world with increasing incidence [27,28,29]. Early diagnosed sporadic CSCC is usually treated with surgery or radiochemotherapy, but the metastatic form is the cause of high mortality and unfavorable prognosis [30,31]. CB1 and CB2 are highly expressed in CSCC and basal cell carcinoma (BCC) [19,24,31] and several studies demonstrated the anti-proliferative and anti-migratory effects of (endo)cannabinoids in these tumors [24,32,33,34,35]. The epithelial-to-mesenchymal transition is a crucial event in tumor progression and metastasis [36,37,38,39]. This process is regulated by a complex signaling network including the Ras/ERK/MAPK and PI3K/AKT/STAT3 pathways responsible for the activation of multiple transcription factors [40,41,42,43,44,45,46,47,48,49]. Growth factors (e.g., EGF, TGF-β, IGF-1), pro-inflammatory cytokines (e.g., TNF-α, IL-6, IL-8), and bioactive lipids also derived from eCB hydrolysis (e.g., arachidonic acid AA, HETEs, prostaglandins) have a role in EMT, inducing the loss of epithelial markers such as E-cadherin and the up-modulation of mesenchymal markers such as N-cadherin, vimentin, fibronectin, and metalloproteinases (MMP-2 and MMP-9) [50,51,52,53,54,55,56]. Based on the ECS’s role in regulating cutaneous homeostasis, the pharmacological modulation of eCBs represents a powerful tool for the management of skin diseases and cancers. A promising therapeutic approach is the use of compounds that act through the inhibition of the eCB degradation enzymes [57,58,59,60], thus excluding most of the side effects observed using exogenous cannabinoids such as sedation, hepatotoxicity, and addiction [61,62]. URB597 is a potent and highly selective FAAH inhibitor that leads to an increase in the levels of AEA and the analogs PEA and OEA, thus enhancing their activity [63]. In the present study, we investigated the anti-tumorigenic effects of URB597 using primary keratinocytes, ex vivo skin explants, and the cutaneous squamous carcinoma cell line A431.

The results demonstrated the ability of URB597 to counteract the EMT process and the migratory capacity by affecting the STAT3 pathway and pro-tumorigenic lipid signaling. These results indicate the need to further explore the ECS and the use of FAAH inhibitors for the development of innovative therapeutic approaches in the treatment of keratinocyte-derived tumors.

## 2. Results

### 2.1. Inhibition of FAAH Induced by URB597 Downregulates Proliferative Capacity and Increases Anandamide Levels in Primary Keratinocytes

To evaluate the effects of URB597 on cell viability and proliferation, normal human keratinocytes (NHKs) were incubated with increasing concentrations of the compound (from 0.01 to 10 μM), and Neutral Red assay and Ki67 immunofluorescence analyses were performed. Cell viability and proliferation were not influenced by URB597 up to a 1 μM dose. A reduction in both parameters started to be evident at 5 μM (Figure 1A,B). To investigate the optimal dose of URB597 able to increase eCB levels, the cells were stimulated with different doses of the molecule, from 0.5 to 10 μM. Intracellular and extracellular concentrations of anandamide (AEA) were analyzed via HPLC. The results revealed a significant increase in intracellular AEA following the treatment with FAAH inhibitor at all of the concentrations evaluated, with the peak being achieved at 1 μM (Figure 1C). The amount of extracellular AEA increased in a dose-dependent manner, reaching a plateau in the dose range between 1 and 10 μM (Figure 1D). To analyze the kinetics of the intracellular accumulation and extracellular release of AEA, NHKs were incubated with URB597 at different doses (0.5–1–5 μM) at the indicated time points. The increase in intracellular AEA was already evident after 3 h of incubation, reached the highest levels at 24 h, and persisted up to 48 h for the 1 μM dose. At URB597 5 μM, the AEA intracellular levels increased after 3 h, decreased at 24 h, and returned to the vehicle-treated cell value at 48 h (Figure 1E). The amount of extracellular AEA significantly increased at 3 h, and was still evident at 24 h, returning to the vehicle-treated cell value at 48 h for all URB597 doses (Figure 1F). Based on these results, a 1 μM dose of URB597 was considered the optimal dose that was able to induce a significant increase in intracellular and extracellular levels of AEA, without inducing cytotoxic effects.

### 2.2. URB597 Contrasts the IGF-1-Induced EMT Process by Regulating Vimentin Expression, E-Cadherin Localization, and Cytokine Release–Involvement of AKT/STAT3 Signaling Pathway

To investigate the role of the URB597-induced increase in eCB levels in counteracting EMT, NHKs were stimulated with IGF-1, a well-known growth factor involved in the switch from an epithelial to mesenchymal phenotype. The immunofluorescence analysis for the expression of the proliferation marker Ki67 showed that, although IGF-1 induced a slight increase in the rate of cell growth, URB597 allowed a return to the proliferation level of the vehicle-treated cells (Figure 2A). Western blot and parallel immunofluorescence analyses demonstrated that URB597 contrasted the increase in vimentin expression induced by IGF-1 (Figure 2B,C). No modification in the overall expression levels of E-cadherin, either in response to IGF-1 or URB597, was observed from the Western blot analysis (Figure 2B). However, the immunofluorescence analysis revealed that the localization of E-cadherin at the cell–cell borders, which was regularly observed in the vehicle-treated keratinocytes, was significantly reduced following IGF-1. Conversely, URB597, alone or in combination with IGF-1, favored the maintenance of a continuous and homogeneous E-cadherin signal at the plasma membrane (Figure 2D). These results were confirmed in ex vivo experiments. Morphological analysis showed that IGF-1 was able to induce epidermal alterations, as evidenced by the presence of intercellular spaces between keratinocytes and increased epidermal thickness. These features were almost completely reverted by URB597 (Figure 2E). Ex vivo samples were immunostained for E-cadherin and vimentin, the latter of which, as expected, identified mesenchymal cells. The IGF-1-treated specimens showed reduced E-cadherin staining, as revealed by a discontinuous signal on the plasma membrane of keratinocytes. URB597 was able to restore a regular and continuous fluorescent pattern mainly localized on the plasma membrane, similar to that detected in the vehicle-treated sample (Figure 2F). To explore whether URB597 was also able to interfere with the production of pro-inflammatory cytokines, the release of IL-6 and IL-8 was assessed in the culture supernatants of NHKs stimulated with IGF-1 via ELISA. In the presence of IGF-1, keratinocytes showed a significant increase in IL-6 and IL-8 production compared to that of vehicle-treated cells. URB597 was able to significantly reduce the extracellular release of both cytokines (Figure 2G).

To investigate the molecular mechanisms by which URB597 regulates the expression of EMT inducers and effectors, we explored the PI3K/AKT and JAK/STAT3 pathways as the main key transducers of IGF-1 signaling. Immunofluorescence analysis performed at different time points demonstrated that the increased phosphorylation levels of AKT and STAT3 in response to IGF-1 were significantly down-modulated by URB597 (Figure 3A,B). The results were also confirmed using parallel Western blot analysis (Figure 3C,D).

### 2.3. URB597 Contrasts the TNF-α-Induced EMT Process by Modulating E-Cadherin Localization and by Promoting Keratinocyte Differentiation

As inflammatory cytokines are known to be involved in favoring the EMT process, we also studied the potential anticancer role of URB597 in counteracting EMT induced by the stimulation with the pro-inflammatory cytokine TNF-α on ex vivo samples. Morphological and E-cadherin/vimentin immunofluorescence analyses showed that the stimulation with TNF-α induced an increase in the intercellular spaces associated with alterations in epidermal cell–cell adhesions, as revealed by the reduced E-cadherin expression, features that were both recovered by URB597 (Figure 4A,B). As TNF-α is known to affect the keratinocyte differentiation process, skin biopsies were evaluated for the expression of keratin 1 (K1) and filaggrin as early and late differentiation markers, respectively. In response to TNF-α stimulation, the keratinocyte differentiation appeared perturbed, showing a reduced expression of K1 at the basal epidermal layers and the presence of a discontinuous signal for filaggrin at the outermost epidermal layers. Following URB597, the staining pattern of both proteins was restored and appeared comparable to those observed in the vehicle-treated samples (Figure 4C).

### 2.4. URB597 Reduces the Proliferative and Migratory Potential and Counteracts the EMT Process in the A431 Squamous Carcinoma Cell Line

To evaluate the anti-tumor effects induced by URB597, we next employed the human cutaneous squamous carcinoma cell line A431 and investigated the ability of the compound to modulate the EMT phenotype. We first analyzed the effect of URB597 on cell viability and growth. The Neutral Red assay revealed no significant differences in cells treated with URB597 at any dose compared with the vehicle (Figure 5A), indicating the higher resistance of tumor cells in comparison to primary keratinocytes. However, the evaluation of cell growth demonstrated a dose-dependent reduction in Ki67-positive cells following URB597, suggesting a possible entry in the G0/G1 resting phase of the cell cycle (Figure 5B). This finding was supported by the parallel increase in the negative regulator of the cell cycle p21 observed in cells treated with URB597 (Figure 5C).

We analyzed the gene expression profile of markers related to EMT (Figure 6A). After URB597 treatment, the results evidenced a significant downregulation of the mRNA expression of the EMT markers FIBRONECTIN, N-CADHERIN, MMP2 and MMP9, SNAI1, and the keratinocyte activation marker KERATIN 6 (K6) (Figure 6A). Western blot and immunofluorescence accompanied by quantitative image analysis revealed the ability of URB597 to downregulate vimentin and fibronectin and up-modulate the keratinocyte differentiation marker involucrin. Conversely, neither the expression nor the localization of E-cadherin varied following the treatment (Figure 6B,C). We next investigated the impact of URB597 on the levels of phosphorylation for AKT, STAT3, p38, and ERK1/2. The results demonstrated that URB597, except for AKT signaling, reduced the phosphorylation levels of all of the signal transducers evaluated (Figure 6D), thus supporting its crucial role in counteracting the activation of the EMT-associated pathways. The reduction in STAT3 phosphorylation was confirmed through immunofluorescence analysis at different time points (Figure 6E). Finally, we investigated the effects of URB597 on cell migration/invasion performing the scratch wound assay on A431 cells stimulated with IGF-1 in the presence or absence of URB597. The quantitation of the edge distance reduction in the different experimental conditions demonstrated the capacity of URB597 to contrast the pro-motogenic effect induced by IGF-1 (Figure 6F).

### 2.5. URB597 Modulates the Profile of Pro-Tumorigenic Lipid Species in A431 Cancer Cells

To assess the effects of URB597 on arachidonic acid (AA) levels and its metabolic pathway, which is known to be involved in EMT promotion, we evaluated the following AA-related bioactive lipids: 5(S)-hydroxyeicosatetraenoic acid (5-HETE) and 15(S)-hydroxyeicosatetraenoic acid (15-HETE) as 5- and 15-lipoxygenase (LOX) metabolites, respectively; 5-hydroperoxyeicosatetraenoic acid (5-HPETE) and 15-hydroperoxyeicosatetraenoic acid (15-HPETE) as precursors and 5-oxoeicosatetraenoic acid (5-oxoETE) and 15-oxoeicosatetraenoic acid (15-oxoETE) as the oxidation products. In addition, we measured PGE2, PGD2, and PGF2-α as representative products of cyclooxygenase 2 (COX-2). The results showed a relevant decrease in free AA following URB597 treatment (Figure 7A), which is directly related to the inhibition of AEA hydrolysis. This reduced availability of AA also resulted in a significant reduction in the levels of its metabolites (Figure 7B–J).

## 3. Discussion

In vitro and in vivo studies demonstrated the therapeutic potential of ECS modulation in different tumors including skin cancers [64,65,66,67]. Although the ECS is involved in tumor physiopathology, the tone of eCBs and the expression levels of their receptors and metabolic enzymes determine the activation of pro- or anti-tumor signaling pathways [11,68]. The pharmacological inhibition of FAAH demonstrated beneficial effects in several cancer cell lines by enhancing the anti-proliferative, pro-apoptotic, and anti-migratory capabilities of eCBs [17,18,57,69,70,71,72].

In the present study, we explored the anti-tumor effect of URB597 treatment in contrasting the EMT process. In NHKs treated with URB597 1 μM, we did not observe a decrease in proliferating cells under basal conditions. We speculate that this is due to the failure to achieve the high concentrations of eCBs (micromolar range) that, when administered exogenously, induce significant anti-proliferative and pro-apoptotic effects [19]. Nevertheless, the same dose of URB597 was able to counteract the growth induced by IGF-1 stimulation. Conversely, in A431 cancer cells, the compound counteracted the proliferative capacity even in the absence of pro-mitogenic stimuli mediated by the growth factors. We hypothesize that the inhibition of FAAH may be even more effective in vivo, where the tumor milieu is enriched with growth factors or cytokines and eCBs reach higher levels. The reduced expression of vimentin underlined the possible role of URB597 not only in disfavoring a crucial effector of EMT, but also in counteracting the pro-tumorigenic signals associated with keratinocyte dedifferentiation. Indeed, it has been reported that in tumor epithelial cells, vimentin modulates keratin expression by switching from a differentiated to a dedifferentiated phenotype [73,74]. The increased migratory potential and loss of cell–cell adhesions associated with the downregulation of E-cadherin expression are other hallmarks of EMT in epithelial cells [75,76]. Immunofluorescence analysis showed that URB597 was able to maintain E-cadherin on the plasma membrane. This localization is fundamental to stabilize cell–cell adhesions and prevent the translocation of β-catenin into the nucleus and the consequent activation of genes involved in cell motility and cancer progression. The investigations on skin explants reinforced these findings, demonstrating the ability of URB597 to ameliorate IGF-1-induced epidermal alterations by reducing enlarged epidermal intercellular spaces and restoring cell–cell adhesions.

In the presence of inflammatory stimuli, which are known to promote and sustain the EMT process, URB597 decreased the expression of vimentin and reduced the extracellular release of IL-6 and IL-8 caused by IGF-1 treatment, thus demonstrating a dual action on both the production of cytokines and their downstream signals. The increased activity of PI3K/AKT and STAT3 signaling pathways has effects on the transcriptional machinery of proto-oncogenes involved in cancer progression and invasiveness [77,78,79,80]. The finding that URB597 reduced the phosphorylation levels of AKT and STAT3 suggests that it may act on multiple signaling pathways that coexist in the complex tumor environment, thus exerting a crucial and broad role in weakening pro-malignant signals. Since few reports have evaluated the role of FAAH inhibitors in counteracting tumor progression in skin cancers, we extended the study to the squamous carcinoma cell line A431. Promising results were obtained in cancer cells, since URB597 reduced proliferative potential, downregulated EMT markers, and promoted keratinocyte differentiation. Cancer cells change lipid metabolism to support tumor progression toward high malignancy and aggressiveness; thus, the identification of lipid-targeted therapeutic strategies is particularly attractive. The tumor milieu is enriched in bioactive lipids, e.g., prostanoids and eicosatetraenoic acids, generated by the arachidonic acid (AA) metabolic pathway, that represent the main drivers of the EMT process [81]. 5(S)-Hydroxyeicosatetraenoic acid (5-HETE) and prostaglandins are metabolites of 5-lipoxygenase (5-LOX) and cyclooxygenase 2 (COX-2) and promote cancer cell proliferation and EMT by inducing the ERK pathway [82,83,84]. eCB degradation enzymes such as FAAH are the primary source of arachidonic acid, free fatty acids, and other metabolites that exert pro-tumorigenic activity. Hence, the reduction in pro-neoplastic signaling lipids (prostaglandins and HETEs) that we observed in response to URB597 treatment suggests a crucial role of this compound in inducing biochemical changes in cancer cells that impair tumor progression and metastasis.

In line with this observation, the inhibition of eCB-degrading enzymes appears to be an intriguing tool for developing innovative anticancer therapies. Reducing the turnover of eCBs prolongs their beneficial anti-tumor effects; however, the levels of FAAH were found to be higher or more downregulated than in healthy tissues depending on the cancer type [85,86,87,88]. This implies that FAAH inhibition has to be contextualized based on the tumor type considered to understand what contribution a decrease in its activity may provide. Data from the literature highlight that the antitumor properties of URB597 are enhanced when the inhibitor is used in combination with AEAs, synthetic analogs, or PEAs due to synergistic action [6,17,19,69,72,89,90]. Brunetti et al. [57] reported that an increased eCB concentration results in the higher activity of PPARs, transcription factors involved in anti-proliferative and anti-inflammatory effects. Therefore, acting simultaneously by inhibiting FAAH and stimulating the activity of PPARs may represent a new approach to cancer therapy.

Further research into the properties of endocannabinoids and the design of multi-target compounds is an emerging field to define new, more potent, and useful pharmacological strategies to reduce the drug doses needed to achieve therapeutic efficacy in skin cancers.

## 4. Materials and Methods

### 4.1. Cell Cultures, Ex Vivo Skin Explants, and Treatments

NHKs were grown in defined Medium 154 supplemented with Human Keratinocyte Growth Supplement (HKGS) (Life Technologies, Monza, Italy), plus antibiotics and calcium chloride (0.07 mM). The cells were subcultured once a week, and the experiments were performed between passages 2 and 4. The A431 cutaneous squamous carcinoma cell line was maintained in basal DMEM (Euroclone, Milan, Italy) supplemented with 10% FBS, L-glutamine (2 mM), and penicillin/streptomycin (100 μg/mL). The cell cultures were routinely tested for Mycoplasma infection. For each experiment, at least three different donors were used.

The NHKs and A431 were maintained in growth-factor-free and serum-free medium, respectively, for 24 h before the treatment with URB597 (cyclohexyl carbamic acid 3′-carbamoylbiphenyl-3-yl ester) dissolved in DMSO (Sigma-Aldrich, Milan, Italy). Control cells were treated with an equal volume of the vehicle. Human Insulin-Like Growth Factor-1 (IGF-1) (Millipore Corporation, Temecula, CA, USA) and Tumor Necrosis Factor-α (TNF-α) (Peprotech, Rocky Hill, NJ, USA) were used.

Ex vivo skin explants after subcutaneous fat excision were cut with a punch of 4 mm diameter and cultured, with the dermis downside, on Transwell permeable supports (6.5 mm diameter, 0.4 μm pore size) into standard well plates at the air–liquid interface. The skin samples were cultured in defined Medium 154 (Life Technologies) supplemented with Human Keratinocyte Growth Supplement (Life Technologies), 10% FBS, antibiotics, and calcium chloride (1.5 mM). After treatments, the skin samples were fixed in formalin and embedded in paraffin.

### 4.2. Neutral Red Assay

Cells treated with URB597 (ranging from 0.01 to 10 μM) for 48 h were then incubated with Neutral Red (0.05 mg/mL) (Sigma Aldrich Srl) for 2 h at 37 °C and lysed in an acetic acid/ethanol solution. The absorbance at 540 nm was measured using a µQUANT spectrophotometer (Biotek Instruments, Winooski, VT, USA). The measurement was performed in triplicate for each sample.

### 4.3. Immunofluorescence Analysis

The NHKs and A431 cells were fixed either with 4% paraformaldehyde followed by 0.1% Triton X-100 to allow permeabilization or with cold methanol at −20 °C. The cells were then incubated with the following primary antibodies: anti-Ki67 rabbit polyclonal antibody (ab15580) (1:300), anti-vimentin rabbit polyclonal antibody (1:400), anti-cytokeratin 1 rabbit polyclonal antibody (ab93652) (1:200) (Abcam) (Cambridge, UK); anti-fibronectin mouse monoclonal antibody (SC-8422) (1:400) (Santa Cruz Biotechnology); anti-E-cadherin mouse monoclonal antibody (1:200) (DakoCytomation); phospho-AKT (Ser473) rabbit monoclonal antibody (1:100), phospho-STAT3 (Tyr705) rabbit monoclonal antibody (1:100) (Cell Signaling). The primary antibodies were visualized using goat anti-rabbit Alexa Fluor 546 conjugate, goat anti-mouse Alexa Fluor 546, and goat anti-mouse Alexa Fluor 488 conjugate antibodies (1:800) (Thermo Fisher Scientific). Coverslips were mounted using ProLong Gold antifade reagent with DAPI (Invitrogen). The fluorescence signal was evaluated by recording the stained images using a CCD camera (Zeiss, Oberkochen, Germany). The quantitative analysis of the fluorescence intensity was performed using the Zen 2.6 (blue edition) software (Zeiss). For the measurement of Ki67 and phospho-STAT3 staining, the number of positive cells/total cells (%) was counted and the results expressed as the mean value ± SD. For E-cadherin, vimentin, fibronectin, and phospho-AKT fluorescent signals, the results were expressed as a fold change of the mean fluorescence intensity/cell ± SD relative to the vehicle-treated cell value, which was set as one by definition. At least 150 cells were measured for each condition from three different experiments.

### 4.4. Immunohistochemical Analysis

Paraffin sections were stained with hematoxylin and eosin (H&E) for histomorphological analysis. For immunofluorescence staining, tissue sections were dewaxed and processed for antigen retrieval and then incubated with the following primary antibodies: anti-E-cadherin mouse monoclonal antibody (M3612) (1:200) (DakoCytomation, Glostrup, Denmark), anti-vimentin rabbit polyclonal antibody (ab92547) (1:400), anti-Filaggrin mouse monoclonal antibody (SC-66192) (1:100) (Santa Cruz Biotechnology, Santa Cruz, CA, USA); anti-Cytokeratin 1 rabbit polyclonal antibody (ab93652) (1:100) (Abcam, Cambridge, UK). Primary antibodies were visualized with goat anti-mouse Alexa Fluor 488 conjugate antibody (1:800) and goat anti-rabbit Alexa Fluor 546 conjugate (1:800) (Thermo Fisher Scientific, Italy). Coverslips were mounted using ProLong Gold Antifade Reagent with DAPI (Life Technologies, Monza, Italy). Staining signals were analyzed by recording images using a CCD camera (Zeiss, Oberkochen, Germany).

### 4.5. Western Blot Analysis

Cells were lysed in RIPA lysis buffer supplemented with a protease/phosphatase inhibitor cocktail (Boster Biological Technology Co., Pleasanton, CA, USA), and then sonicated. The total cell lysates were clarified using centrifugation at 12,000 rpm for 10 min at 4 °C and then stored at −80 °C until analysis. Following the spectrophotometric protein measurement, equal amounts of protein were resolved on acrylamide SDS-PAGE and transferred onto nitrocellulose membrane (Amersham Biosciences, Milan, Italy). The protein transfer efficiency was checked with Ponceau S staining (Sigma-Aldrich, St. Louis, MO, USA). The membranes were first washed with water, blocked with EveryBlot Blocking Buffer (Bio-Rad Laboratories Srl, Milan, Italy) for 10 min at room temperature, and then treated overnight at 4 °C with primary antibodies (according to the data sheet instructions). The primary antibodies anti-vimentin (ab92547) (1:1000) and anti-involucrin (ab53112) (1:500) (Abcam, Cambridge, UK); anti-E-cadherin (M3612) (1:1000) (DakoCytomation (Glostrup, Denmark); anti-Fibronectin (SC-8422) (1:1000) (Santa Cruz Biotechnology, CA, USA); anti-phospho-AKT (Ser473) (#4060) (1:1000), anti-AKT (#2920) (1:1000), anti-phospho-STAT3 (Tyr705) (#9145) (1:2000), anti-STAT3 (#9139) (1:2000), anti-phospho-p38 MAP kinase (Thr180/Tyr182) (#4511) (1:1000), anti-p38 MAPK (#9212) (1:1000), anti-phospho-p44/42 MAPK (ERK1/2) (Thr202/Tyr204) (#4370) (1:2000), anti-ERK2 (#1647) (1:1000), anti-p21 Waf1/Cip1 (#2947) (1:1000) were purchased from Cell Signaling (Danvers, MA, USA). Secondary antibodies: anti-mouse IgG HRP-conjugated antibody (#7076) (1:3000) and anti-rabbit IgG HRP-conjugated antibody (#7074) (1:8000) (Cell Signaling, Danvers, MA, USA). The antibody complexes were visualized using Amersham ECL Western Blotting Detection Reagent (GE Healthcare, Buckinghamshire, UK). Anti-α-actin antibody (A5441) (1:10,000) and anti-GAPDH antibody (G9545) (1:5000) (Sigma-Aldrich) were used as the loading control. The protein levels were quantified by measuring the optical densities of specific bands using the UVI-TEC Imaging System (Cambridge, UK). The control value was taken as one-fold in each case.

### 4.6. Protein Determination Using Sandwich Enzyme-Linked Immunosorbent Assay (ELISA)

Culture supernatants were collected and centrifuged to remove cell detritus. Aliquots were stored at −80 °C until use. The IL-6 and IL-8 protein levels were determined using commercially available ELISA kits (Diaclone SAS, Besancon Cedex, France) according to the manufacturer’s instructions. The results were normalized for the number of cells contained in each sample. The measurement was performed in duplicate for each sample. The absorbance at 450 nm was recorded using a DTX880 Multimode Detector spectrophotometer (Beckman Coulter srl., Milan, Italy).

### 4.7. RNA Extraction and Real-Time RT-PCR

Total RNA was isolated using the AurumTM Total RNA Mini kit (Bio-Rad, Milan, Italy) according to the manufacturer’s instructions. The total RNA samples were stored at −80 °C until use. Following DNAse I treatment, cDNA was synthesized using a mix of oligo-dT and random primers and RevertAidTM First Strand cDNA synthesis kit (Thermo Fisher Scientific, Monza, Italy) according to the manufacturer’s instructions. Real-time RT-PCR was performed in a total volume of 10 μL with SYBR Green PCR Master Mix (Bio-Rad, Milan, Italy) and 200 nM concentration of each primer. The sequences of all primers used are shown in Table 1. Reactions were carried out in triplicate using a CFx96TM Real-Time System (Bio-Rad). Melt curve analysis was performed to confirm the specificity of the amplified products. The expression of mRNA (relative) was normalized to the expression of GAPDH mRNA using the change in the Δcycle threshold (ΔCt) method and calculated based on 2^−ΔCt^.

### 4.8. Scratch Assay

A431 cells were seeded on 35 mm Petri dishes and allowed to grow until confluence. A cell-free area was then generated by wounding the cell monolayer using a pipette tip. After repeated washes, the cells were immediately fixed (time zero, T0) or treated with IGF-1 (100 ng/mL) in the presence or absence of URB597 (1 μM) and then fixed after 24 h. Images were recorded using a CCD camera and the migratory ability was determined by measuring the edge distance using the Zen 2.6 software. The results are expressed as fold change with respect to T0, which was set as 100 by definition from three experiments.

### 4.9. Lipid Extraction

Lipids were extracted accordingly using the Bligh and Dyer procedure with slight modifications [91]. Briefly, the lipids were extracted with chloroform:methanol (2:1) (2 × 1 mL) after the addition of butylhydroxytoluene to prevent the oxidation of oxygen-sensitive compounds. Then, 10 µL of a mixture of deuterated standards (d17C16:0 8µM; d4PGD2 1µM; d4PGE2 1 µM; d85HETE 1 µM; d815HETE 1 µM) were added to control the analytical performance and to calculate the relative abundance of the lipid species detected. The samples were vortexed and centrifuged at 10,000× *g* for 10 min at 4 °C. The extraction procedure was repeated twice and the organic layers were collected and evaporated under nitrogen. The dried lipid extract was stored at −80 °C until the analysis.

### 4.10. AEA High-Performance Liquid Chromatography/Mass Spectrometry

HPLC–MS/MS analyses were performed on a quadrupole tandem mass spectrometer 6400 coupled to a 1200 HPLC Liquid Chromatography system from Agilent Technologies (Palo Alto, CA, USA). A Zorbax Eclipse XDB C8 column (150 × 4.6 i.d. 5 µm particle size) from Agilent Technologies (Palo Alto, CA, USA) was used and operated at 40 °C. Isocratic elution (30% mobile phase A and 70% of mobile phase B) at a constant flow rate of 0.45 mL/min was performed. Mobile phase A consisted of 2 mM ammonium acetate in water containing 0.1% formic acid (*w*/*v*) and mobile phase B consisted of acetonitrile containing 0,1% formic acid (*w*/*v*). The autosampler was thermostated at 5 °C. Sample aliquots of 5 µL were injected into the analytical system. HPLC-MS/MS analyses were carried out in positive electrospray ionization mode (ESI^+^). Quantification was performed with multiple reaction monitoring (MRM) of the transition m/z 348 → 62 for AEA and 356 → 62 for d8-AEA (IS for AEA). The collision energy was set at 12 eV. For the quantification of AEA, a linear curve was generated where the ratio of AEA standard peak area to d8-AEA peak area was plotted versus the amount of AEA standard. The results were calculated as pmoles/10^6^ cells and reported as fold change in comparison to vehicle-treated cells.

### 4.11. Assessment of Pro-Inflammatory Lipid Mediators by High-Performance Liquid Chromatography/Mass Spectrometry

The prostaglandins (PGD2, PGE2, PGF2α) and lipoxygenase-derived metabolites of arachidonic acid (AA), in particular, 5HETE, 5oxoETE, 5HpETE, 15HETE, 15oxoHETE, and 15 HpETE, in A431 cells were measured via HPLC–MS/MS. Chromatographic separation was carried out using the Agilent Technologies 1200 HPLC Liquid Chromatography System (Palo Alto, CA, USA) with a C18 column (Symmetry, 3.5 μm, 100 mm × 2.1 mm, Waters) as previously reported [92,93]. Briefly, the mobile phase flow rate was 0.2 mL/min. Mobile phase A consisted of acetonitrile–water–formic acid (20:80:0.1, *v*/*v*/*v*), and mobile phase B consisted of acetonitrile–formic acid (100:0.1, *v*/*v*). Mobile phase B was increased from 0 to 100% in a linear gradient over 6 min and maintained at 100% until 10 min. Mobile phase B was then decreased to 0% from 10 to 11 min and maintained at 0% until 22 min. The column temperature was maintained at 40 °C. The injection volume was 5 μL. The overall run time was 25 min. Negative ion electrospray tandem mass spectrometry was carried out with an Agilent Technologies triple quadrupole 6400 mass spectrometer at unit resolution with multiple reaction monitoring (MRM) performed by monitoring the following transitions: PGD2 351 → 233; PGD2-d4 (IS for PGD2) 355 → 233; PGE2 351 → 315; PGE2-d4 (IS for PGE2) 355 → 319; PGF2α353 → 193; PGF2α-d4 (IS for PGF2α) 357 → 197; 5HETE 319 → 115; 5oxoETE 317 → 245; 5HpETE 335 → 203; 5HETE-d8 (IS for 5HETE, 5oxoETE and 5HpETE) 327 → 116; 15HETE 319 → 175; 15oxoETE 317 → 113; 15HpETE 335 → 113; 15HETE-d8 (IS for 15HETE, 15oxoETE and 15HpETE) 327 → 226; AA 17 303 → 303; d17C16:0 (IS for AA) 272 → 272. For each compound to be quantified, an internal standard was selected and a linear curve was generated where the ratio of analyte standard peak area to internal standard peak area was plotted versus the amount of analyte standard. The results were calculated as nM/mg of total proteins.

### 4.12. Statistical Analysis

Data were represented as mean ± standard deviation (SD) of a minimum of three independent experiments. Statistical significance was assessed using paired Student’s *t*-test or ANOVA followed by Tukey’s multiple comparison test using GraphPad Prism (GraphPad Prism Software 8.0.2, Boston, MA, USA). *p* < 0.05 was considered statistically significant.

## Figures and Tables

**Figure 1 ijms-24-17379-f001:**
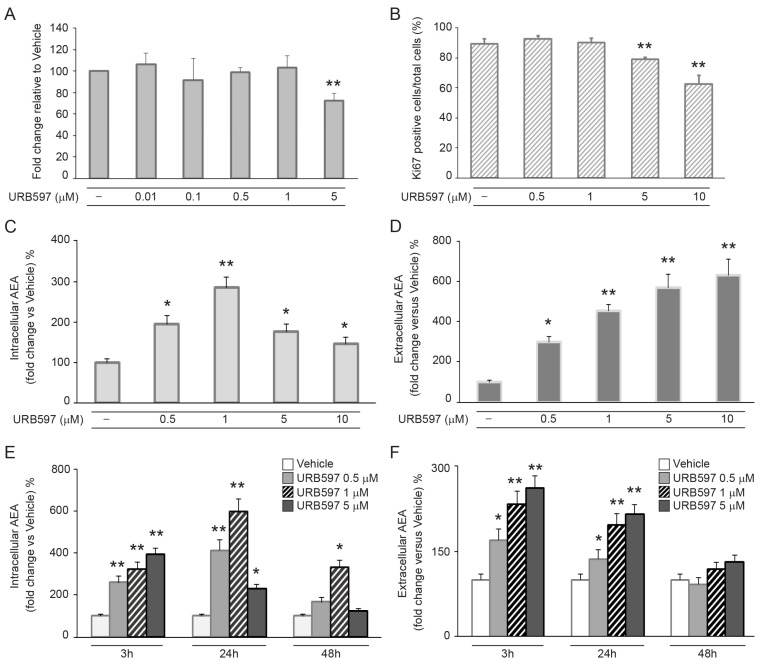
URB597 downregulates proliferative capacity and increases anandamide levels in NHKs. (**A**) Cell viability evaluated by Neutral Red assay on NHKs treated with increasing concentrations of URB597 for 48 h. Results are expressed as fold change ± SD relative to vehicle-treated cell value, which was set as 100 by definition. (**B**) Quantitative analysis of Ki67-positive cells assessed through immunofluorescence in response to URB597 treatment for 24 h. Results are expressed as the mean value ± SD of positive cells/total cells (%). Intracellular AEA production (**C**) and extracellular release (**D**) in response to increasing concentrations of URB597 after 24 h. Results are expressed as fold change ± SD relative to vehicle-treated cell value, which was set as 100 by definition. Intracellular (**E**) and extracellular (**F**) amounts of AEA at different time points of incubation with increasing doses of URB597. Vehicle (white bar), URB597 0.5–1 and 5 μM (light grey, striped bar and dark grey bar, respectively). Results are expressed as fold change ± SD relative to vehicle-treated cell value, which was set as 100 by definition. * *p* < 0.05 vs. vehicle-treated cells; ** *p* < 0.01 vs. vehicle-treated cells.

**Figure 2 ijms-24-17379-f002:**
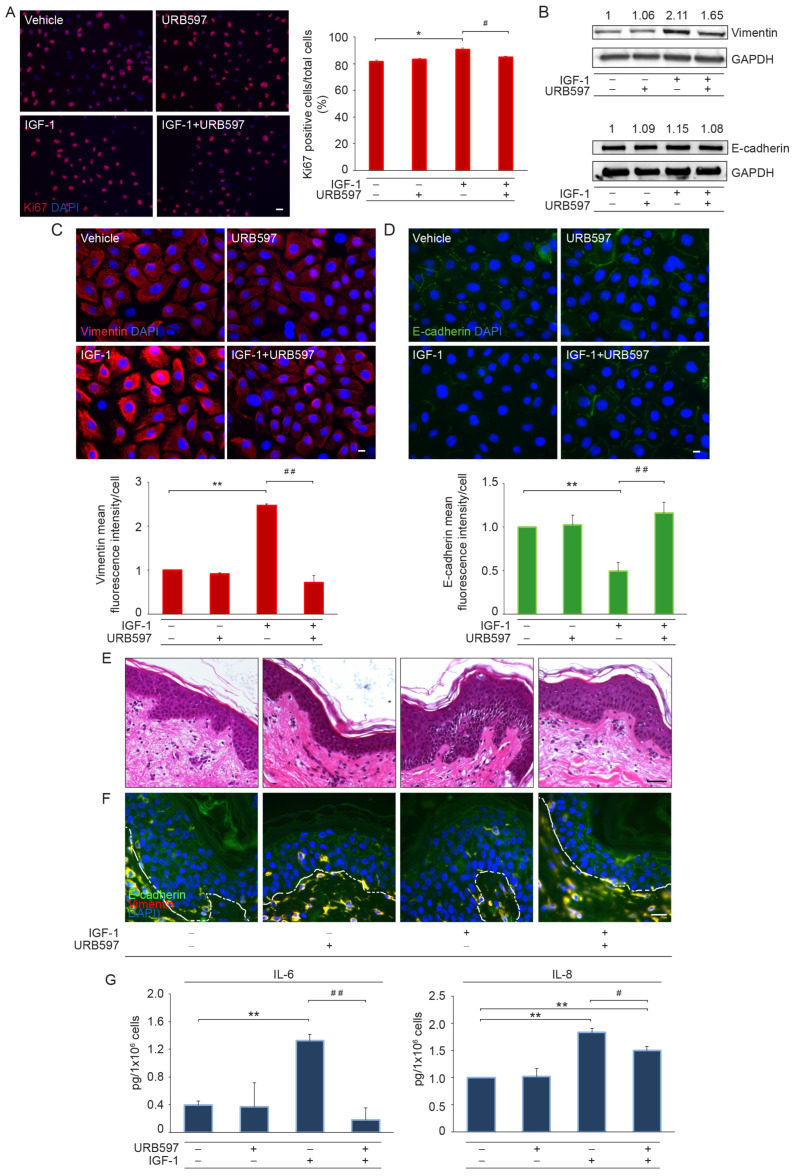
URB597 contrasts the EMT phenotype induced by IGF-1 by reducing vimentin expression, modulating E-cadherin localization, and decreasing the release of pro-inflammatory cytokines. (**A**) Immunofluorescence and quantitative analysis of Ki67 stained NHKs following IGF-1 (100 ng/mL) stimulation for 48 h in the presence or absence of URB597 (1 μM). Results are expressed as the mean value ± SD of positive cells/total cells (%). Scale bar: 20 μm. (**B**) Western blot of vimentin and E-cadherin on NHKs in response to IGF-1 and URB597. Representative blots are shown. GAPDH was used as an endogenous loading control. Densitometric scanning of band intensities was performed to quantify the change in protein expression. Results are expressed as fold change ± SD relative to vehicle-treated cell value, which was set as 1-fold in each case. Immunofluorescence and signal intensity measurement of vimentin (**C**) and E-cadherin (**D**) in NHKs treated as above. Nuclei are counterstained in DAPI. Scale bar: 10 μm. Data are representative of three independent experiments. (**E**) Hematoxylin-and-eosin-stained sections and (**F**) immunofluorescence for E-cadherin and vimentin proteins from ex vivo skin explants stimulated with IGF-1 in the presence or absence of URB597 (5 μM). Nuclei in (**F**) are counterstained with DAPI and the basal membrane is outlined with a white dashed line. Scale bar: 50 μm (**E**), 20 μm (**F**). (**G**) IL-6 and IL-8 quantitation by ELISA in URB597- and vehicle-treated cells following IGF-1 stimulation for 48 h. * *p* < 0.05 vs. vehicle-treated cells; ** *p* < 0.01 vs. vehicle-treated cells; ^#^ *p* < 0.05 vs. IGF-1-treated cells; ^##^ *p* < 0.01 vs. IGF-1-treated cells.

**Figure 3 ijms-24-17379-f003:**
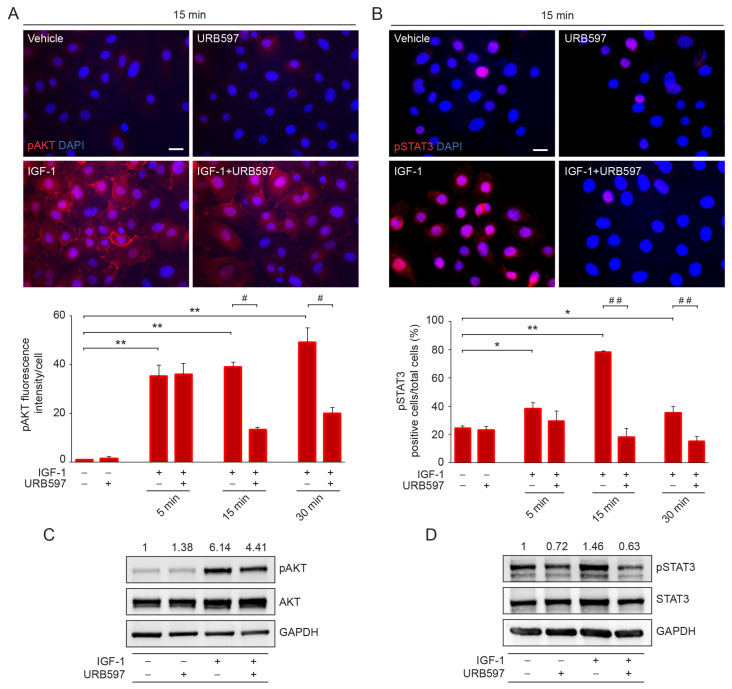
URB597 affects AKT/STAT3 signal pathways by reducing their phosphorylation levels. (**A**) Immunofluorescence and quantitative analysis of the mean fluorescence intensity of pAKT in IGF-1 (100 ng/mL)-stimulated cells following the treatment with URB597 (1 μM) at the indicated time points. Results are expressed as fold change ± SD of the intensity mean/cell relative to the vehicle-treated cell value, which was set as 1 by definition. (**B**) Immunofluorescence and quantitative analysis of positive pSTAT3 cells/total cells following IGF-1 treatment in the presence or absence of URB597 at the indicated time points. Results are expressed as positive cells/total cells ± SD (%). Scale bar: 20 μm. Western blot of pAKT (**C**) and pSTAT3 (**D**) in NHKs stimulated with IGF-1 in the presence or absence of URB597. Representative blots are shown. GAPDH was used as an endogenous loading control. Densitometric scanning of band intensities was performed to quantify the change in protein expression. Results are expressed as fold change ± SD relative to vehicle-treated cell value, which was set as 1-fold in each case. * *p* < 0.05 vs. vehicle-treated cells; ** *p* < 0.01 vs. vehicle-treated cells; ^#^ *p* < 0.05 vs. IGF-1-treated cells; ^##^ *p* < 0.01 vs. IGF-1-treated cells.

**Figure 4 ijms-24-17379-f004:**
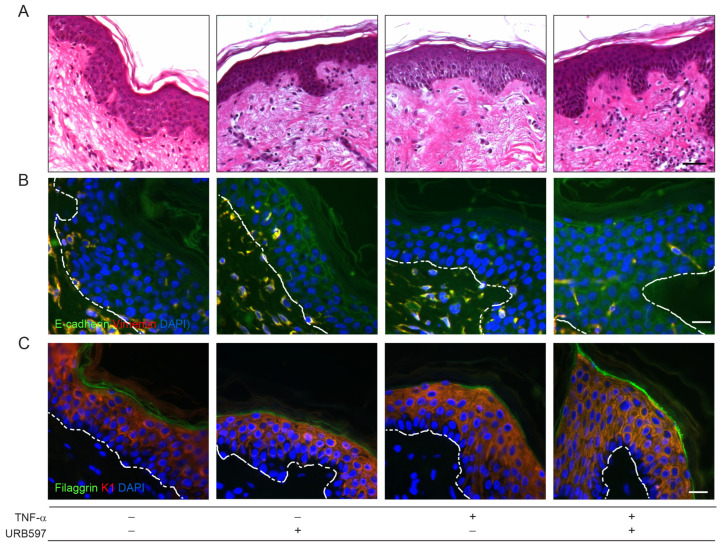
URB597 counteracts the EMT process triggered by TNF-α modulating E-cadherin localization and promoting keratinocyte differentiation. (**A**) Hematoxylin-and-eosin-stained sections from vehicle and TNF-α (20 ng/mL) stimulated ex vivo skin explants treated or not with URB597 (5 μM) for 4 days. Double immunofluorescence staining for E-cadherin and vimentin (**B**) and filaggrin and K1 (**C**) on ex vivo skin specimens treated as above. Nuclei are counterstained with DAPI and the basal membrane is outlined with a white dashed line. Scale bar: 50 μm (**A**), 20 μm (**B**,**C**).

**Figure 5 ijms-24-17379-f005:**
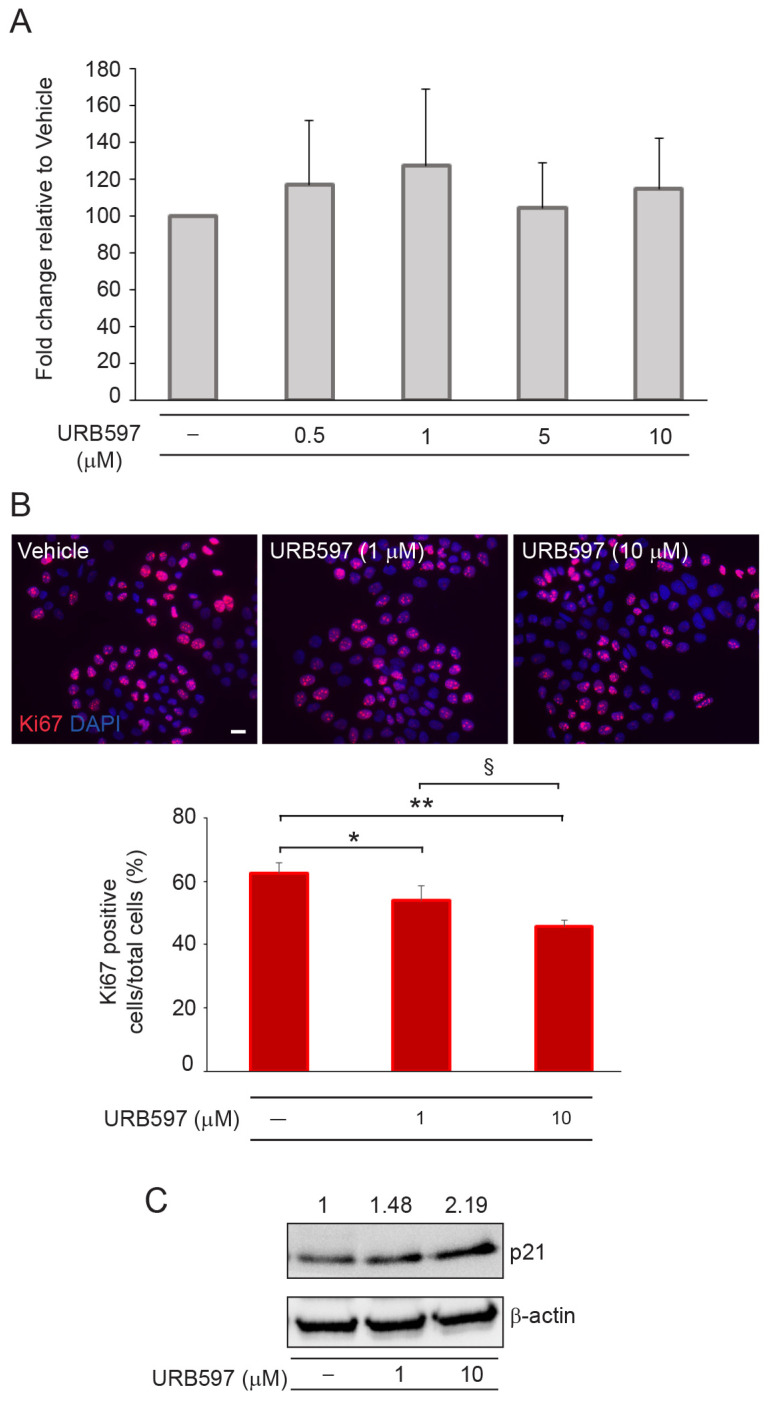
Effects of URB597 on cell viability and proliferative potential of A431 squamous carcinoma cell line. (**A**) Cell viability evaluated using Neutral Red assay on A431 cells treated with increasing concentrations of URB597 for 48 h. Results are expressed as fold change ± SD relative to vehicle-treated cell value, which was set as 100 by definition. (**B**) Immunofluorescence and quantitative analysis of Ki67-positive cells in response to 1 μM and 10 μM URB597 treatment. Results are expressed as the mean value ± SD of positive cells/total cells (%). Nuclei are counterstained with DAPI. Scale bar: 20 μm. (**C**) Western blot of p21 in A431 cells treated with 1 μM and 10 μM doses of URB597 in the presence or absence of URB597. Representative blots are shown. β-actin was used as an endogenous loading control. Densitometric scanning of band intensities was performed to quantify the change in protein expression. Results are expressed as fold change ± SD relative to vehicle-treated cell value, which was set as 1-fold in each case. * *p* < 0.05 vs. vehicle-treated cells; ** *p* < 0.01 vs. vehicle-treated cells; ^§^ *p* < 0.05 vs. 1 μM URB597-treated cells.

**Figure 6 ijms-24-17379-f006:**
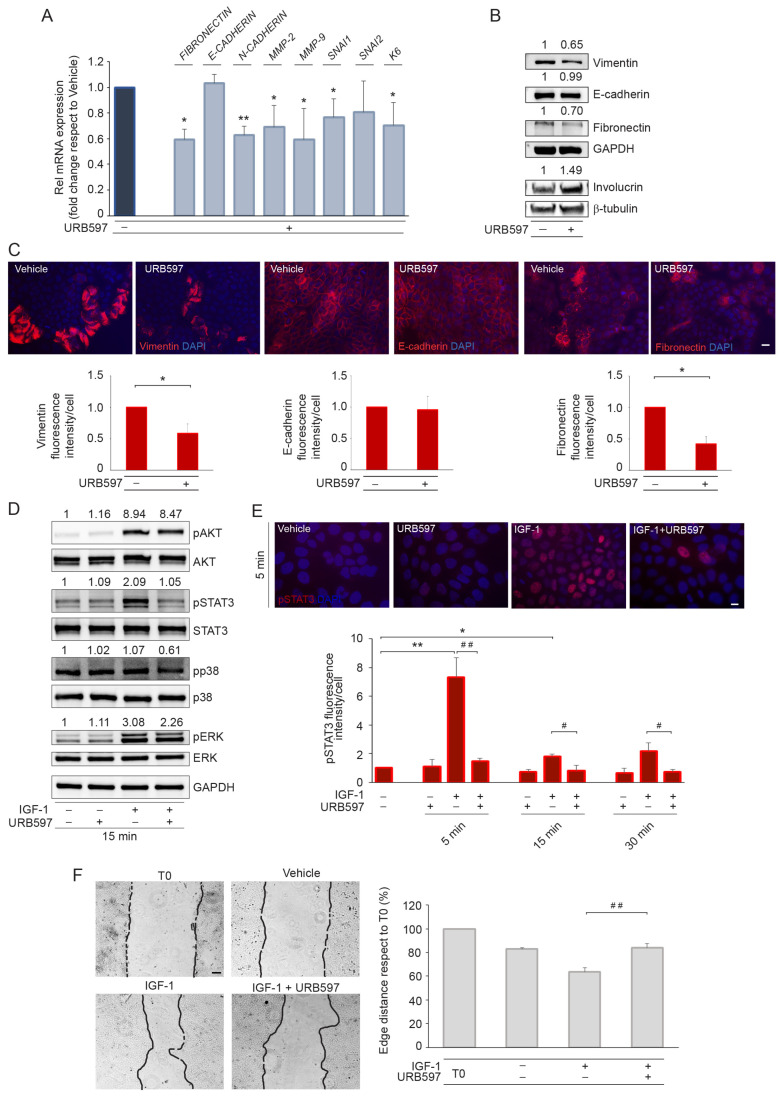
URB597 counteracts the EMT process by downregulating EMT markers and associated intracellular signaling pathways and decreases migratory potential in A431 cells. (**A**) Real-time RT-PCR analysis of a panel of genes involved in EMT phenotype on A431 cells treated with URB597 (1 μM). All mRNA values were normalized against the expression of GAPDH and were expressed relative to vehicle-treated cell value. (**B**) Western blot and corresponding densitometric analysis of vimentin, E-cadherin, fibronectin, and involucrin on A431 cell lysates treated with URB597 for 48 h. Representative blots are shown. GAPDH and β-tubulin were used as endogenous loading controls. Densitometric scanning of band intensities was performed to quantify the change in protein expression. Results are expressed as fold change ± SD relative to vehicle-treated cell value, which was set as 1-fold in each case. (**C**) Immunofluorescence staining and signal intensity measurement of vimentin, E-cadherin, and fibronectin on A431 cells in the presence or absence of URB597 treatment. Results are expressed as mean intensity ± SD/cell in comparison to vehicle-treated cell value, which was set as 1 by definition. Nuclei are counterstained with DAPI. Scale bar: 20 μm. (**D**) Western blot and corresponding densitometric analysis of IGF-1 (100 ng/mL)-induced phospho-AKT, phospho-STAT3, phospho-p38, and phospho-ERK on A431 cells in the presence or absence of URB597. Representative blots are shown. (**E**) Immunofluorescence and quantitative analysis of the mean fluorescence intensity ± SD/cell of pSTAT3 in IGF-1-stimulated A431 following the treatment with URB597 at 5, 15, and 30 min. Nuclei are counterstained with DAPI. Scale bar: 10 μm. (**F**) Representative images of the scratch wound assay on A431 cells stimulated with IGF-1 in the presence or absence of URB597 for 24 h. Quantitative analysis of the reduction in the distance of the leading edges with respect to the T0 value, which was set as 100 for definition. Scale bar: 100 μm. * *p* < 0.05 vs. vehicle-treated cells; ** *p* < 0.01 vs. vehicle-treated cells; ^#^ *p* < 0.05 vs. IGF-1-treated cells; ^##^ *p* < 0.01 vs. IGF-1-treated cells.

**Figure 7 ijms-24-17379-f007:**
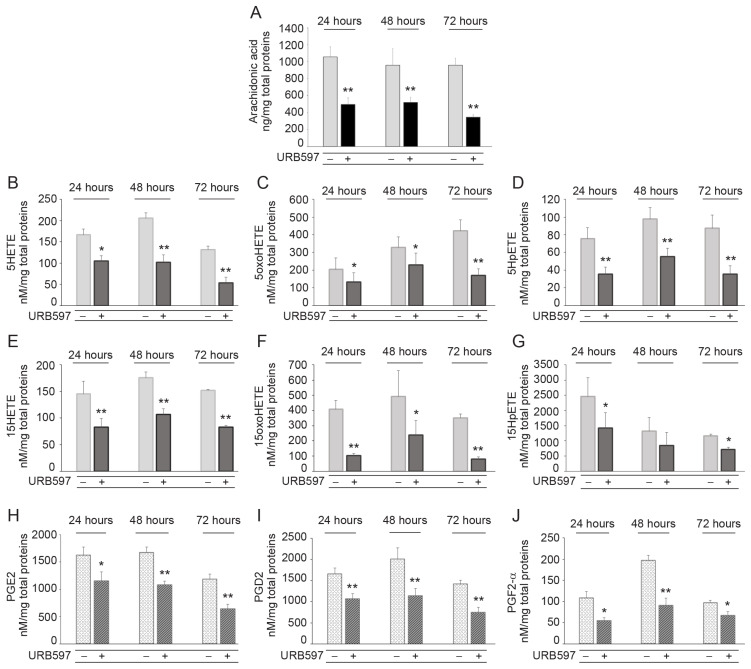
URB597 modifies the composition of pro-tumorigenic lipid species in A431 cells. Quantification via LC-MS/MS of intracellular production of arachidonic acid AA (**A**), 5HETE (**B**), 5oxoHETE (**C**), 5HpETE (**D**), 15HETE (**E**), 15oxoHETE (**F**), and 15HpETE (**G**) in A431 cells at different time points of incubation with URB597 (1 μM). Intracellular amount of PGE2 (**H**), PGD2 (**I**), and PGF2-alpha (**J**) in A431 cells following the treatment with URB597 at the indicated time points. Lipid amount is measured as nM/mg total proteins and the results are expressed as mean value ± SD. * *p* < 0.05 vs. vehicle-treated cells; ** *p* < 0.01 vs. vehicle-treated cells.

**Table 1 ijms-24-17379-t001:** Forward (F) and reverse (R) primers used for the real-time RT-PCR analysis.

Gene	Oligonucleotide Sequences (5′-3′)	Amplicon Size	Accession Number
*E-CADHERIN*	F: GAACGCATTGCCACATACAC R: ATTCGGGCTTGTTGTCATTC	118 bp	NM_004360.5
*FIBRONECTIN*	F: CCTCGAAGAGCAAGAGGCAG R: GCTTCAGGTTTACTCTCGCA	202 bp	NM_001365522.2
*GAPDH*	F: TGCACCACCAACTGCTTAGC R: GGCATGGACTGTGGTCATGAG	198 bp	NM_001289746
*K6*	F: AGTCCTGCTTCTCTTC R: CTGCTGTGGCTCCTGATG	107 bp	NM_005554.4
*MMP-2*	F: AGAAGGCTGTGTTCTTTGCAG R: AGGCTGGTCAGTGGCTTG	88 bp	NM_004530.6
*MMP-9*	F: TGACAGCGACAAGAAGTG R: CAGTGAAGCGGTACATAGG	143 bp	NM_004994.3
*N-CADHERIN*	F: GGACTATGATTACCTGAACGACTG R: AGTTAAAGCCTAGCTTCTGAATGC	161 bp	NM_001792.5
*SNAI1*	F: ACTATGCCGCGCTCTTTCC R: GTCGTAGGGCTGCTGGAAG	111 bp	NM_005985.4
*SNAI2*	F: TGGTTGCTTCAAGGACACAT R: GCAAATGCTCTGTTGCAGTG	77 bp	NM_003068.5

## Data Availability

The data that support the findings of this study are available on request from the corresponding author.

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
