# Peer review of "Targeting Fatty Acid Amide Hydrolase Counteracts the Epithelial-to-Mesenchymal Transition in Keratinocyte-Derived Tumors"

_ijms, 2023, doi:10.3390/ijms242417379_

Round 1
Reviewer 1 Report
Comments and Suggestions for Authors
The authors describe the issues in a very comprehensive way. The tests are performed using various research methods, e.g. western blot. This allows for a more detailed description of the problem and emphasizes the veracity of the results obtained. The only comment is the quality of the figures, especially the font in the axis signatures is too small and you have to guess why their quality is low. However, the topic of treatment and the search for new methods of treatment is very important nowadays, when cancer is a disease of civilization. After correcting the figures, the article is ready for publication.
Author Response
Response to Reviewer 1 Comments
Comment:
The authors describe the issues in a very comprehensive way. The tests are performed using various research methods, e.g. western blot. This allows for a more detailed description of the problem and emphasizes the veracity of the results obtained. The only comment is the quality of the figures, especially the font in the axis signatures is too small and you have to guess why their quality is low. However, the topic of treatment and the search for new methods of treatment is very important nowadays, when cancer is a disease of civilization. After correcting the figures, the article is ready for publication.
Response:
We thank the Reviewer for the positive comments on our study. We apologize for the poor quality of the figures, probably due to the conversion of the original file containing Figures in Tiff format to PDF version.
As suggested, we increased the font of the axes in the graphs and improved the resolution of the images by directly uploading the PDF.
In addition, we removed the doses of the treatments inside the Figures and we added them to the legends.

Reviewer 2 Report
Comments and Suggestions for Authors
Kovacs et al aims to investigate the anticancer property of URB597, a fatty acid amide hydrolase (FAAH) inhibitor, particularly by examining its ability to counteract EMT. To this end, the authors examine the effect of URB597 on proliferation of NHK cells as well as anandamide levels. The authors then check the counteracting effect of URB597 on IFG-1-mediated EMT. Kovacs et al then probe into the potential role of ATK/STAT3 pathway in NHK cells. They further perform experiments on A431 cells to examine the effect of URB597 on migration and related signal transduction pathways. Additionally, the authors interrogate the pro-tumorigenic lipid composition in A431 cells upon URB597 treatments. They report that URB597 is capable of modulating the EMT process by downregulation mesenchymal markers and by reducing migration. The authors also provide evidence that these phenotypes are likely to come about through AKT/STAT3 pathways.
I believe that the authors provide nice and convincing evidence on URB597-mediated targeting of FAAH that could be used to counteract the EMT in keratinocyte-derived tumors. The experiments are well-designed and conducted and the manuscript is well-organized, making it easy to follow. I sugges the following minor corrections before warranting publication.
Minor points
1. A new paragraph (indentation) in lines 58 and 84.
2. Line 94, please write out “NHK” in its first use.
3. Many examples of “ex vivo” throughout the manuscript that should be italicized.
4. The resolution of Figure 1 (especially 1A, but preferably all panels) should be improved.
5. Line 170, “hematoxylin & eosin” not “hematoxylin&eosin
6. Line 481, the sentence should begin with a capital letter.
Comments on the Quality of English LanguageAlmost fine.
Author Response
Response to Reviewer 2 Comments
Comment:
Kovacs et al aims to investigate the anticancer property of URB597, a fatty acid amide hydrolase (FAAH) inhibitor, particularly by examining its ability to counteract EMT. To this end, the authors examine the effect of URB597 on proliferation of NHK cells as well as anandamide levels. The authors then check the counteracting effect of URB597 on IFG-1-mediated EMT. Kovacs et al then probe into the potential role of ATK/STAT3 pathway in NHK cells. They further perform experiments on A431 cells to examine the effect of URB597 on migration and related signal transduction pathways. Additionally, the authors interrogate the pro-tumorigenic lipid composition in A431 cells upon URB597 treatments. They report that URB597 is capable of modulating the EMT process by downregulation mesenchymal markers and by reducing migration. The authors also provide evidence that these phenotypes are likely to come about through AKT/STAT3 pathways.
I believe that the authors provide nice and convincing evidence on URB597-mediated targeting of FAAH that could be used to counteract the EMT in keratinocyte-derived tumors. The experiments are well-designed and conducted and the manuscript is well-organized, making it easy to follow. I sugges the following minor corrections before warranting publication.
Response:
We thank the Reviewer for appreciating our manuscript and for the suggested corrections.
Accordingly, we modified the text and the changes have been highlighted.
We apologize for the low quality of the figures. As suggest, we improved their resolution. We also removed the doses of the treatments from the Figures and we added them to the legends.
